# Characterization of SiO$_2$/4H-SiC Interfaces in 4H-SiC MOSFETs: A Review

**Patrick Fiorenza *** ⓘ**, Filippo Giannazzo**ⓘ **and Fabrizio Roccaforte**ⓘ

Consiglio Nazionale delle Ricerche–Istituto per la Microelettronica e Microsistemi (CNR-IMM), 95121 Catania, Italy; filippo.giannazzo@imm.cnr.it (F.G.); fabrizio.roccaforte@imm.cnr.it (F.R.)
* Correspondence: patrick.fiorenza@imm.cnr.it

**Abstract:** This paper gives an overview on some state-of-the-art characterization methods of SiO$_2$/4H-SiC interfaces in metal oxide semiconductor field effect transistors (MOSFETS). In particular, the work compares the benefits and drawbacks of different techniques to assess the physical parameters describing the electronic properties and the current transport at the SiO$_2$/SiC interfaces (interface states, channel mobility, trapping phenomena, etc.). First, the most common electrical characterization techniques of SiO$_2$/SiC interfaces are presented (e.g., capacitance- and current-voltage techniques, transient capacitance, and current measurements). Then, examples of electrical characterizations at the nanoscale (by scanning probe microscopy techniques) are given, to get insights on the homogeneity of the SiO$_2$/SiC interface and the local interfacial doping effects occurring upon annealing. The trapping effects occurring in SiO$_2$/4H-SiC MOS systems are elucidated using advanced capacitance and current measurements as a function of time. In particular, these measurements give information on the density (~$10^{11}$ cm$^{-2}$) of near interface oxide traps (NIOTs) present inside the SiO$_2$ layer and their position with respect to the interface with SiC (at about 1–2 nm). Finally, it will be shown that a comparison of the electrical data with advanced structural and chemical characterization methods makes it possible to ascribe the NIOTs to the presence of a sub-stoichiometric SiO$_x$ layer at the interface.

**Keywords:** 4H-SiC; MOSFET; trapping states; electrical characterization; nanoscale characterization

---

## 1. Introduction

Silicon carbide (4H-SiC) is the best candidate to replace silicon in power electronics applications. In particular, its high critical electric field and large band gap make it possible to design devices with a high breakdown voltage (BV), having specific on-resistance ($R_{on,sp}$) two orders of magnitude lower than silicon-powered devices. This concept is clearly illustrated in the $R_{on,sp}$ versus BV plot depicted in Figure 1.

The $R_{on,sp}$ versus BV plot shows that planar metal oxide semiconductor field effect transistors (MOSFETs) have approached the 4H-SiC unipolar limit for BV values larger than 1200 V. However, both the commercial and the R&D devices designed for operating in the 600–900 V range [1,2] are still far from the ideal unipolar limit. This behavior can be explained by looking at the structure of the planar power MOSFETs and at the SiO$_2$/SiC interface.

Figure 2 shows the schematic cross section of the elementary cell of the planar power MOSFETs, indicating also the vertical current path connecting source to drain electrodes in the on-state and each resistance components. As can be seen, the total $R_{on,sp}$ of the device can be written as the sum of different contributions [3]:

$$R_{on,sp} = R_{ch} + R_a + R_{JFET} + R_{drift} + R_{sub},$$ (1)

where $R_{ch}$ is the channel resistance, $R_a$ is the accumulation (between the n-type epitaxy and the insulator) region resistance, $R_{JFET}$ is the resistance of the JFET (junction Field-Effect Transistor) region (from the surface to the end of the body region), $R_{drift}$ is the resistance of the drift region after taking into account current spreading from the JFET region (from the body region to the end of the epitaxial layer), and $R_{sub}$ is the resistance of the n-type doped substrate.

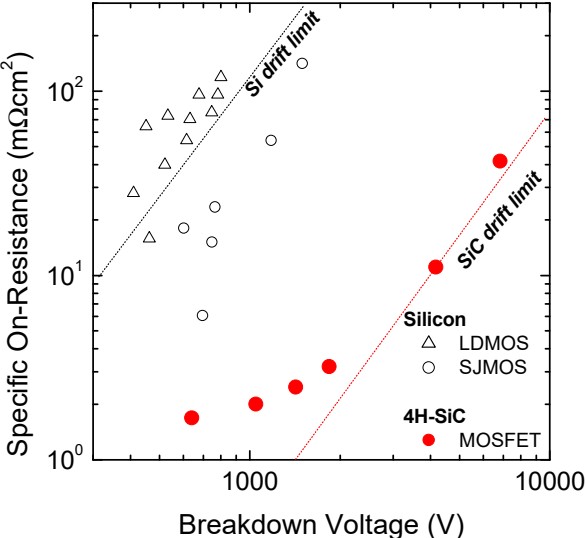

**Figure 1.** Comparison of $R_{on,sp}$ as a function of the breakdown voltage breakdown voltage (BV) for Si and 4H-SiC power metal oxide semiconductor field effect transistors (MOSFETs). The solid lines are the theoretical unipolar limit. The experimental data are taken from Reference [2].

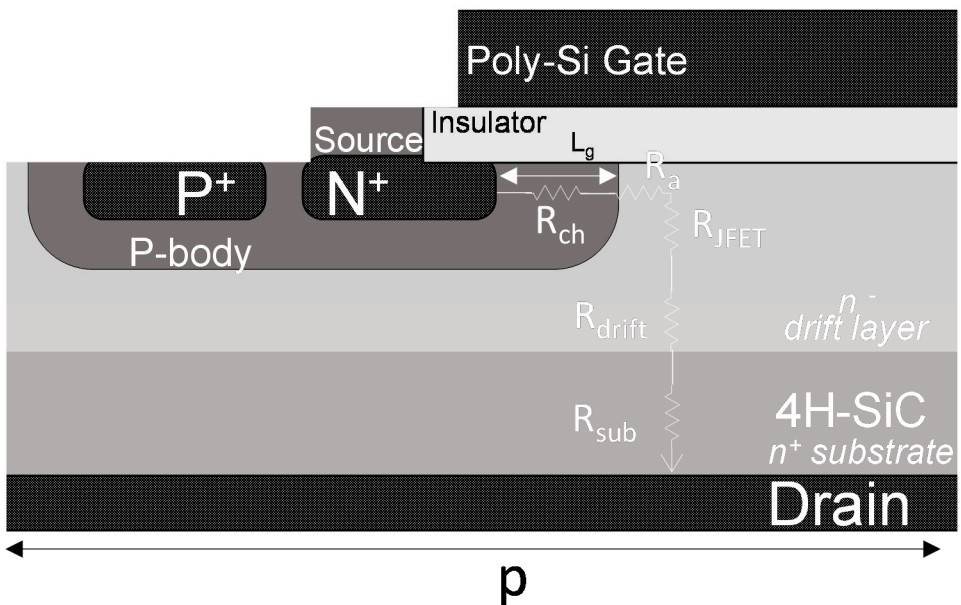

**Figure 2.** Cross section of a planar 4H-SiC power MOSFET. The vertical path of the current from source to drain electrode, as well as the resistive contributions, are also indicated.

While $R_a$ and $R_{JFET}$ can be minimized by appropriately scaling the device layout, and $R_{sub}$ can be reduced by thinning the substrate, the control of the channel resistance contribution $R_{ch}$ is more critical. In fact, the channel resistance contribution $R_{ch}$ is given by [3]:

$$R_{ch} = \frac{(L_{ch} \cdot p)}{\mu_{inv} C_{ox}(V_G - V_{th})} \tag{2}$$

where $p$ is the pitch of the MOSFET elementary cell, $L_{ch}$ is the channel length, $\mu_{inv}$ is the mobility for electrons in the channel (inversion layer), $C_{ox}$ is the specific capacitance of the gate oxide, $V_{th}$ is the threshold voltage, and $V_G$ is the applied gate bias. It is important to emphasize that power devices must have a good trade-off between the ON- and OFF-state. In particular, for a given breakdown voltage and threshold voltage, the choice of a drift layer and body region with fixed doping levels and thicknesses are required. Hence, the minimum pitch size is obtained, avoiding the overlap of the depleted region of the body-drain p-n diode (in the JFET region). Furthermore, the $R_{ch}$ can be influenced by the $C_{ox}$. However, to guarantee the reliability of power devices, insulators with high critical breakdown field, large band gap, and high melting point are desired. So far, the most robust gate insulation solution is the $SiO_2$, but recent literature is exploring alternative solutions, such as $Al_2O_3$ [4] and AlON [5]. Moreover, the channel resistance $R_{ch}$ can be lowered by reducing the channel length. However, when the MOSFET operates in interdiction and it is reverse biased, short-channel phenomena have to be avoided. Hence, the reduction of $R_{ch}$ is strictly related to the optimization of the inversion channel mobility $\mu_{inv}$. This optimization clearly requires a good comprehension of the physical phenomena governing the transport in the channel. In addition, besides the inversion channel mobility $\mu_{inv}$, the threshold voltage $V_{th}$ is another important parameter, which directly influences the channel resistance. Hence, channel mobility and threshold voltage must be accurately controlled to optimize the device performance and to fully exploit the benefits of the 4H-SiC material.

Several review papers recently reported on the physical and technological issues which limit the performances and reliability of 4H-SiC MOSFETs in power electronics applications [6–8]. In general, 4H-SiC MOSFETs are characterized by a low inversion channel mobility and the occurrence of $V_{th}$ instability phenomena under bias stress [9,10]. The $V_{th}$ stability is a very important request, to avoid degradation and/or irreversible device failure under long time stress condition at temperatures above 150 °C.

The $SiO_2$/4H-SiC metal-oxide-semiconductor (MOS) system is the most important part of the transistor, and it is schematically depicted in Figure 3. Clearly, the behavior of the SiC power MOSFET depends critically on the properties of the $SiO_2$/4H-SiC MOS system. In particular, some relevant regions can be identified in the $SiO_2$/4H-SiC MOS system. The first one is the $SiO_2$/4H-SiC interface, which is characterized by the presence of a distribution of interface states ($D_{it}$), close either to the valence or conduction band edge [11,12]. Then, a second region inside the gate insulator is characterized by the presence of slow near interface oxide traps (NIOTs) and bulk traps [13,14]. Finally, a "modified" 4H-SiC region close to the interface with $SiO_2$ is typically present in the MOS system. This latter region may be different under the electrical or structural/chemical point of view with respect to the bulk 4H-SiC semiconductor [15].

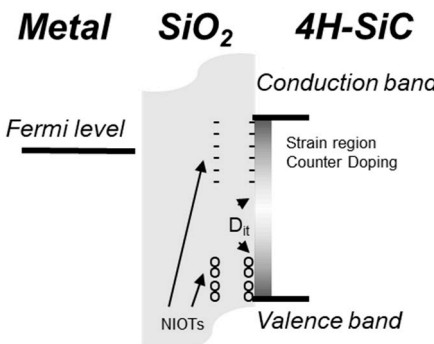

**Figure 3.** Graphical representation of the $SiO_2$/4H-SiC interface, indicating the regions of interest and interface states and oxide traps.

Clearly, in such a complex scenario, only the cross correlation of a variety of characterization techniques can make it possible to obtain an exhaustive picture of the $SiO_2$/SiC interface and, hence, to provide useful feedback for the 4H-SiC MOSFET manufacturers.

The goal of this paper is to give an overview of the current issues related to the characterization of SiO$_2$/4H-SiC interfaces. In particular, the importance of correlating conventional electrical analyses of devices and test patterns, with more advanced characterization techniques at the nanoscale, is highlighted. As an example, it will be shown that using nanoscale approaches can make it possible to get insights on the spatial homogeneity of the surface potential at the SiO$_2$/4H-SiC interface and explain the deviations from the ideal behaviour. Finally, the relevance of time-resolved measurements to study trapping states at the SiO$_2$/4H-SiC interface will be also emphasized.

## 2. Interface States Density and Channel Mobility in 4H-SiC MOS-Based Devices

In general, the parameter used to describe the channel behaviour in 4H-SiC MOSFETs is the field effect mobility, $\mu_{FE}$, determined from the device transconductance in the linear region according to equation:

$$\mu_{FE} = \frac{L}{WC_{ox}V_{DS}} \frac{\partial I_D}{\partial V_{GS}} \tag{3}$$

where $W$ is the channel width, $V_{DS}$ is the source-drain potential, and $\partial I_D / \partial V_{GS}$ is the MOSFET transconductance. The field effect mobility contains the physical information on the modulation of the channel conductivity by the application of the gate bias. Typically, $\mu_{FE}$ is determined in lateral MOSFETs, i.e., where the device resistance is given only by the contacts and the channel contributions. In fact, in a lateral MOSFET, the JFET, accumulation, and drift resistance contributions are absent.

The SiO$_2$/4H-SiC interface is characterized by the presence of a relatively high density of interface states $D_{it}$ (> $10^{12}$ eV$^{-1}$cm$^{-2}$) close to the conduction band edge. It is widely accepted that the presence of a large amount of interface states $D_{it}$ is detrimental for the field effect mobility $\mu_{FE}$ and on-resistance R$_{ON}$ of 4H-SiC MOSFETs [7,16]. According to the studies reported in the last decades, the $\mu_{FE}$ can be improved with different passivation processes of the SiO$_2$/SiC. The passivation can be achieved by thermal annealing in nitrogen-rich atmospheres (NO or N$_2$O) [17–20]. Table 1 compares several literature data concerning the channel mobility measured in 4H-SiC MOSFETs, fabricated either on epitaxial or ion-implanted layers, or MOS capacitors, and subjected to annealing of the gate oxides in nitrogen-rich conditions or combined oxidation (including high temperature Argon annealing) [21–30] that reduced the $D_{it}$ amount. Typically, the passivation annealing temperatures are in the range 1100–1400 °C. As can be seen, a notable increase of the channel mobility (up to 20–50 cm$^2$V$^{-1}$s$^{-1}$) can be achieved upon nitridations of the gate oxide, with respect to the values typically measured in non-annealed thermal oxides (typically below 5 cm$^2$V$^{-1}$s$^{-1}$) [19]. The improvement of the channel mobility is often, but not always, accompanied by a reduction of the interface state density $D_{it}$ (down to the low $10^{12}$ eV$^{-1}$cm$^{-2}$ range). It is assumed that the passivation mechanism of SiO$_2$/4H-SiC interface consists in the elimination of an excess of carbon at the interface or in the saturation of dangling bonds [31].

Rozen et al. [28] reported a correlation between the MOSFET channel mobility, the amount of nitrogen incorporated at the SiO$_2$/4H-SiC interface upon annealing in NO, and the interface charged states $N_{it}$ (the integral of $D_{it}$ over the gap). Besides the NO or N$_2$O post-annealing treatments, other processes can be beneficial for $D_{it}$ and/or $\mu_{FE}$. As an example, Kobayashi et al. [32] reported that a high temperature annealing (up to 1500 °C) in Ar can improve the SiO$_2$/4H-SiC interface, by avoiding the formation of fast states still present after NO or N$_2$O processes. Hatakeyama et al. [33] showed a dependence of the $D_{it}$ on the annealing time in NO at 1250 °C, obtaining the best mobility results after 60 min annealing. More recently, Asaba et al. [34] combined a low temperature (900 °C) annealing in O$_2$ to promote the subsequent Nitrogen incorporation at the interface with a N$_2$ annealing at 1300 °C, reaching a field effect mobility of 50 cm$^2$V$^{-1}$s$^{-1}$.

**Table 1.** Summary of field effect mobility $\mu_{FE}$ and interface states density $D_{it}$ data in 4H-SiC MOSFETs and MOS capacitors. The processing conditions of the gate oxide (on epitaxial or implanted body), the extraction method of $D_{it}$, and the doping $N_A$ of the p-type body concentration are also reported.

| Process | Temperature (°C) | $\mu$ (cm$^2$V$^{-1}$s$^{-1}$) | $D_{it}$ (cm$^{-2}$eV$^{-1}$) | Methods | $N_A$ (cm$^{-3}$) | Ref. |
|---|---|---|---|---|---|---|
| Dry | 1175 | 4 | $9 \times 10^{12}$ | $G_p/\omega$ | $1 \times 10^{16}$ epi | [19,21] |
| NO | 1175 | 32 | N.A. | N.A. | $8 \times 10^{15}$ epi | [20,25] |
| NO 10 min | 1250 | 20 | $2 \times 10^{14}$ | Hall | $1.3 \times 10^{15}$ | [33] |
| NO 60 min | 1250 | 38 | $8 \times 10^{13}$ | Hall | $2.3 \times 10^{15}$ | [28,33] |
| NO 120 min | 1250 | 34 | $5-6 \times 10^{13}$ | Hall | $2.7 \times 10^{15}$ | [33] |
| $N_2O$ | 1300 | 55-99 | $4 \times 10^{11}$ | $G_p/\omega$ | $1 \times 10^{16}$ epi | [19,24] |
| $N_2O$ | 1150 | 24-40 | $4-8 \times 10^{11}$ | $G_p/\omega$ | $1 \times 10^{17}$ imp | [29,30] |
| $N_2O$ | 1410 | N.A. | $1 \times 10^{12}$ | High-low | $5 \times 10^{15}$ epi | [26] |
| $O_2 + N_2$ | 900 + 1300 | 50 | $3 \times 10^{11}$ | High-low | $5 \times 10^{15}$ epi | [34] |
| Ar | 1500 | N.A. | $2 \times 10^{12}$ | C–$\psi$ | $N_D = 1 \times 10^{16}$ epi | [22,32] |

It is clear from the data collected in Table 1 that a correlation between the channel mobility $\mu_{FE}$ and the $D_{it}$ is not straightforward, because the $\mu_{FE}$ and $D_{it}$ are typically determined using different techniques (high-low method, conductance method $G_p/\omega$, C–$\psi$ method, etc.). Tables 2 and 3 summarize the most widely used characterization techniques to determine channel mobility and interface states, illustrating their advantages and limitations.

The conventional characterization methods used to determine the $D_{it}$ at SiO$_2$/4H-SiC interfaces are the high-low method [35] and the frequency dependent parallel conductance measurements ($G_p/\omega$ or conductance method). The high-low method compares a low-frequency C–V curve with a high-frequency C–V curve. The high frequency C–V curve is acquired at a frequency (e.g., 1 MHz) where the interface traps are supposed to not respond to the AC signal. Low frequency means that interface traps and minority carrier inversion charges should respond to the measurement AC probe frequency. Unfortunately, standard C–V performed at 1 MHz on 4H-SiC do not fully satisfy this condition. On the other hand, the $G_p/\omega$ (or conductance method) is sensitive to $D_{it}$ in the portion of the band gap that corresponds to depletion and weak inversion, and also the capture cross-sections for majority carriers, and surface potential fluctuations. The $G_p/\omega$ method measures the equivalent parallel conductance $G_p$ of an MOS capacitor as a function of frequency and bias. However, in SiO$_2$/4H-SiC interfaces there is a non-negligible amount of $D_{it}$ above the Fermi level, where the $G_p/\omega$ method is not sensitive. Hence, the abovementioned techniques underestimate the distribution of $D_{it}$ when $q^2 \cdot D_{it} > C_{ox}$ [36]. Furthermore, the $G_p/\omega$ method is not sensitive to the fast states that can be undetectable in the usually employed frequency range (up to a few MHz; see Table 2) [16]. Thus, in order to detect the fast states, Yoshioka et al. [37] proposed a novel method based on the difference between the theoretical and quasi-static capacitances in SiC MOS capacitors (C–$\psi$). In fact, by employing the C–$\psi$ method, it was possible to make a correct determination of the $D_{it}$ and, hence, to establish a correlation between the $D_{it}$ and the field effect mobility $\mu_{FE}$. [38], as shown in Figure 5.

Another characterization technique, the charge pumping method, commonly used in silicon devices [39], is recently rising relevance also for SiC devices. This method is based on measuring the transistor base charge-pumping current while applying voltage pulses (variable amplitudes and lengths) to its gate. In particular, the difficulty in interpreting anomalous data collected on 4H-SiC MOSFETs has been firstly explained by Okamoto et al [40] in terms of geometric components and the acceptor-like interface states. In fact, due to the low channel mobility, the charge carriers need a certain time to cross the channel. Hence, an accurate setting of the experimental procedure is needed. More recently, Salinaro et al. [41] found an appropriate temperature, frequency, and bias amplitude to demonstrate not only the interface state distribution but also a not homogeneous doping distribution in the channel due to the device processing.

**Table 2.** Summary of the most widely used characterization techniques for estimating the interface state density $D_{it}$.

| Characterisation Methods | Device Typology | Advantages | Limitations | $D_{it}$ Range (cm$^{-2}$eV$^{-1}$) | Ref. |
|---|---|---|---|---|---|
| High-low | MOS | Precise in devices with low $D_{it}$ values | It can be affected by insulator traps | $10^{10}$–$10^{12}$ | [35] |
| $G_p/\omega$ | MOS/ MOSFETs | Able to separate the frequency response of slow and fast states | Unable to probe fast states at frequency >100MHz | $>1 \times 10^9$ | [35] |
| C-ψ | MOS/ MOSFETs | Accurate on large $D_{it}$ value and on a broad frequency range | Need of a precise knowledge of the doping | $>1 \times 10^{11}$ | [37] |
| Sub-threshold | MOSFET | Quick method to qualitatively compare different processes | Difficult to estimate the absolute $D_{it}$ value | $>2 \times 10^{11}$ | [35] |
| Charge pumping | MOSFET | Access to acceptor and $D_{it}$ states | Difficult estimation of $D_{it}$ near the conduction band edge | $>1 \times 10^{11}$ | [40,41] |

**Table 3.** Summary of the most widely used characterization techniques to evaluate the mobility in 4H-SiC MOSFETs.

| Characterisation Methods | Device Typology | Advantages | Disadvantages | Reference |
|---|---|---|---|---|
| Effective mobility ($\mu_{eff} = \frac{L}{WQ_n} \frac{\partial I_D}{\partial V_{DS}}$) Field effect mobility ($\mu_{FE} = \frac{L}{WC_{ox}V_{DS}} \frac{\partial I_D}{\partial V_{GS}}$) | Lateral MOSFET | Fast comparison of different processes. Wafer level characterization | Incorrect estimation of the amount of the free carriers | [35] |
| Hall effect | MOSFET Hall bars | Separation of the trapped and free carriers | Need for multi terminal structures. Characterization on discrete devices only | [33,34] |

Although the 4H-SiC MOSFET mobility is often determined from Equation (3), this approach would give the correct channel mobility value only in the absence of charge trapping effects. In fact, under this assumption, the total density of free electrons $n_{free}$ contributing to the channel conduction corresponds to the total inversion layer electrons density $n_{TOT}$ ($n_{free} = n_{TOT}$). However, in the presence of electron trapping at interface states, a substantial fraction of the inversion layer electrons is trapped ($n_{free} < n_{TOT}$) [42]. Hence, using the device transconductance (from the current-voltage characteristics) leads to an underestimation of the actual channel mobility.

On the other hand, the effective mobility, $\mu_{eff}$, is extracted from the device conductance in the linear region (typically at $V_{DS}$ = 50–100 mV), according to the equation:

$$\mu_{eff} = \frac{L}{WQ_n} \frac{\partial I_D}{\partial V_{DS}} \tag{4}$$

where $Q_n$ is charge density the channel (C/cm$^2$).

However, the measure of $\mu_{eff}$ presents some weaknesses. First, the effective charge in the inversion layer $Q_n = C_{ox}(V_{GS} - V_{th})$ is difficult to measure accurately. In fact, to accurately determine the $Q_n$, complicated experimental setups are needed in order to perform the split $C$–$V$ measurements [35] separating the source-drain and gate-base capacitance response. Second, the additional series capacitances introduced by the interface states can influence the mobility estimation.

Generally, in 4H-SiC MOSFETs, the value of the effective mobility $\mu_{eff}$ is lower than the field effect mobility $\mu_{FE}$. This discrepancy is due mainly to the presence of interface traps affecting the explicit $Q_n$ dependence of $\mu_{eff}$ ($\mu_{FE}$ is not explicitly related to $Q_n$) but not draining conductance or draining current. In fact, $Q_n$ is the sum $n_{free} + n_{Trap}$. Since $n_{Trap}$ is not contributing to the channel conduction, $Q_n$ is larger than $n_{free}$, thus resulting in a considerable underestimation of $\mu_{eff}$ ($< \mu_{FE}$).

A different way to characterize the properties of the MOSFET channel is determining the Hall Effect Mobility, $\mu_{Hall}$. In fact, the Hall measurement gives a direct measurement of the free electrons $n_{free}$. Hence, the big advantage of the Hall measurement is that $\mu_{Hall}$ and $n_{free}$ are determined independently. Consequently, the technique provides correct mobility results, independent of the charge trapping [42]. On the other hand, the total amount of charge at each gate bias value—or, alternatively, at each surface potential value—aligns with $Q_n = C_{ox}(V_{GS}-V_{th})$ and this allows the extraction of the interface states profile. The values of the Hall mobility are higher than the field effective mobility. Obviously, for Hall measurements, special test patterns must be designed (i.e., Hall bars in the MOSFET channel). Moreover, the accurateness of this method is correlated to the knowledge of the Hall scattering factor [43]. Recently, Hall measurements have been used by Hatakeyama et al. [33] to determine the total amount of the trapped and free carriers in the channel of 4H-SiC MOSFETs subjected to different NO treatments, and their results are reported in Table 1. They have demonstrated that the improvement of the $D_{it}$ and of the Hall mobility is not strictly related to the duration of the NO treatments. In fact, a post-oxidation-annealing (POA) duration that exceeds a certain time can produce a detrimental effect compared with a shorter POA duration.

After the mentioned clarifications on the experimental methods to determine both the interface state density and the MOSFET channel mobility, it is possible to try to draw a correlation between the $\mu_{FE}$ and the $D_{it}$.

Figure 4 shows the data obtained from Nakazawa et al. [38] correlating the $\mu_{FE}$ and the total amount of the interface states $N_{it}$ (the energy integral of the $D_{it}$). As can be seen, unlike the conventional characterization methods (e.g., 1 MHz conductance method and high-low), a nice correlation of the peak mobility $\mu_{FE}$ with the reverse of the interface trap density ($1/N_{it}$) is visible when the C-$\psi$ method is used for the quantification of $D_{it}$. Clearly, the values of the mobility increase, decreasing the amount of $N_{it}$. The mentioned dependence is a strong indication that a Coulomb-scattering contributes to limiting the carriers transport at the SiO$_2$/SiC interface [29].

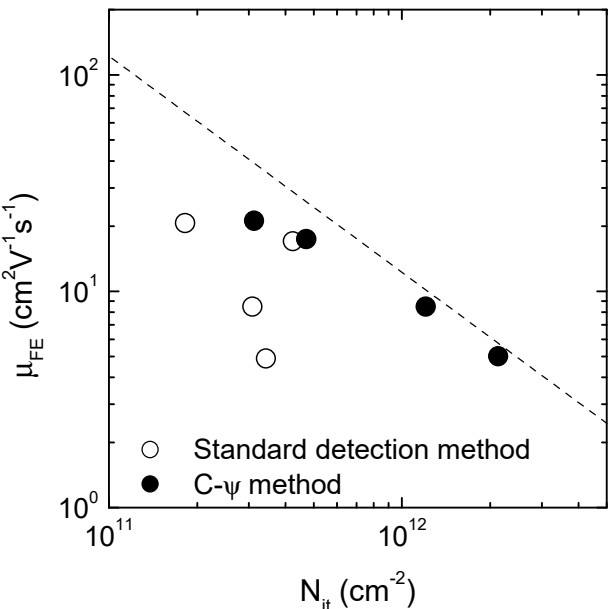

**Figure 4.** Experimental field effect mobility $\mu_{FE}$ values as a function of the integral of the interface state density ($N_{it}$) evaluated with different methods. A clear correlation is obtained when $N_{it}$ is evaluated using the C-$\psi$ method. The data were taken from Reference [38].

Very recently, Hauck et al. [44] presented an analytical method that overcomes some of the limits of many methodologies originally developed for silicon devices. In particular, their analytical model makes it possible to correct the underestimation of the charge carrier concentration and mobility.

The method provides a formulation of the three-terminal 4H-SiC MOSFETs characterization of any geometry, consisting of an accurate determination of device parameters hampered by the presence of traps at the interface. They parametrized the SiC/SiO$_2$-specific interface trap spectrum, including the body of known data. The resulting parameters, such as the mobility, the charge carrier density, and threshold voltage, have been demonstrated to be in good agreement with Hall effect measurements.

Clearly, the characterization of interface states at SiO$_2$/4H-SiC interfaces is continuously an object of scientific debate. In this context, to overcome some limitations of the conventional electrical characterization techniques, the cross correlation with nanoscale electrical characterization techniques is mandatory to obtain a better view of the SiO$_2$/SiC interface physics.

## 3. Nanoscale Electrical Properties of SiO$_2$/4H-SiC Interfaces

### 3.1. Electrical Characterization

The electrical behaviour of 4H-SiC MOSFETs is affected by the microscopic nature of the SiO$_2$/SiC inversion channel. As an example, the roughness of the channel region, often characterized by a typical "step bunching" of the surface, can have an impact on both the interface state and mobility of the fabricated devices.

In this context, some authors reported on an anisotropy of the channel mobility $\mu_{FE}$ in 4H-SiC MOSFETs, with the channel in different orientations [30,45]. In particular, 4H-SiC MOSFETs fabricated with the channel along the [1–100] direction (along the bunched steps) exhibited a higher channel mobility compared to those fabricated with channel along the [11–20] direction (across the bunched steps) [45]. Frazzetto et al. [29] explained this effect, taking into consideration the impact of both $D_{it}$ and surface roughness in the scattering contributions to the field effect mobility. However, the conventional device electrical characterization methods were not able to give an exhaustive picture of the involved physical phenomena.

Hence, in recent years, strong efforts have been devoted to an accurate characterization of SiO$_2$/4H-SiC interfaces, employing nanoscale electrical/structural analyses of the interface (e.g., scanning probe microscopy, transmission electron microscopy).

As an example, Figure 5 compares the dispersion of the flat band voltage $\Delta V_{FB}$ measured both by conventional $C$–$V$ measurements and by scanning capacitance microscopy (SCM).

As can be seen (Figure 5a), the $C$–$V$ characterization of 25 large area state-of-the-art (N$_2$O annealed) MOS capacitors (i.e., $100 \times 100$ μm$^2$) gives a narrow dispersion of the flat band voltage, indicating a homogeneous SiO$_2$/4H-SiC interface behaviour. Interestingly, nanoscale electrical measurements can provide information that cannot be assessed using macroscopic MOS capacitors. The SCM measures the capacitance variations ($dC/dV$) produced by the modulating bias at each atomic force microscope (AFM) tip position [30]. The SCM tip is scanned in contact mode on the bare surface of a semiconductor (e.g., 4H-SiC) coated by an insulating film (e.g., SiO$_2$), thus forming a nanometric tip-insulator-semiconductor (nanoMOS) device. The sample is biased by a DC bias and a high frequency (between 10 and 100 kHz) AC bias. The capacitance sensor connected to the probe detects the capacitance variations induced by the modulating AC bias in the tip-insulator-semiconductor (nanoMOS) structure. Due to the nanometric contact size, the capacitance of the tip-insulator-semiconductor nanoMIS structure is extremely small, to the order of $\sim 10^{-18}$ F (aF). To disentangle this small capacitance signal from the large stray capacitance values (associated with the cantilever, wires, etc.), a lock-in amplifier connected to the capacitance sensor selects the capacitance contributions at the AC bias modulating frequency. As a result, the lock-in amplifier output is an arbitrary unit signal proportional to the differential capacitance ($dC/dV$) of the nanoMOS system. As matter of fact, the $dC/dV$ peak values are primarily related to the local flat band voltage in the region underneath the tip. Hence, the spatial variation of the $dC/dV$ peak positions can be correlated to $V_{FB}$ dispersion.

Figure 5b shows the $V_{FB}$ dispersion obtained by the $dC/dV$ SCM signal collected at 25 points—a matrix $5 \times 5$ at 1 μm distance in the XY directions—on the same sample. Evidently, the $V_{FB}$ dispersion

collected at nanoscale (Figure 5b) is broader than that collected on macroscopic capacitors (Figure 5a), thus suggesting the presence of a non-homogeneous $D_{it}$ distribution.

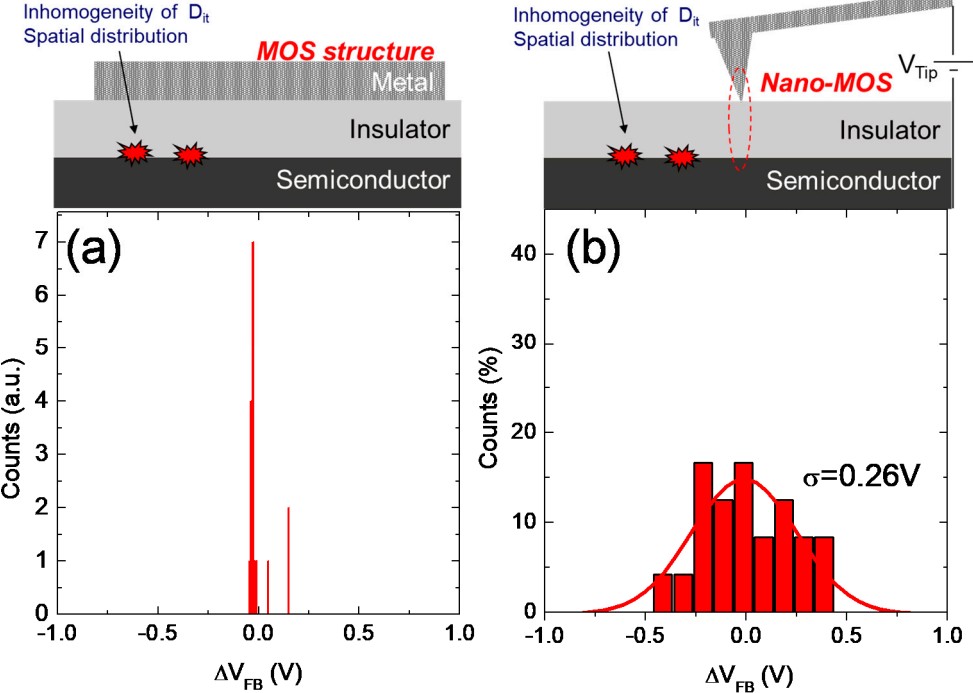

**Figure 5.** Dispersion of the flat band voltage shift $\Delta V_{FB}$ obtained from $C-V$ measurements on 25 macroscopic MOS capacitors on a 6″ wafer (**a**), and by scanning capacitance microscopy (SCM) measurements on arrays of 25 tip positions (with 1 μm spacing) (**b**).

A possible origin of the non-homogeneous distribution of the electrical properties of the $SiO_2/4H$-SiC interface can be identified by comparing the spatial distribution of the capacitance signal collected using the SCM.

Figure 6a,b show the morphology and the capacitance maps collected on the bare surface of a nitridated $SiO_2/4H$-SiC sample at 1150 °C in $N_2O$. The non-uniform spatial distribution of the capacitance signal can be associated with the spatial fluctuation of the surface potential. For example, in a faceted surface, the non-uniform $D_{it}$ spatial distribution is correlated with [46] the different contributions to the total $D_{it}$ value given by the $(11-2n)$ planes of the surface facets and the $(0001)$ basal planes [47]. Saitoh et al. [48] reported on the variation in the density of the interface states in MOS capacitors fabricated on 4H-SiC with different miscut angles, moving from 8°, i.e., the $(0001)$ largest basal plane, up to 90° toward the $[11-20]$ direction, i.e., the $(11-20)$ plane.

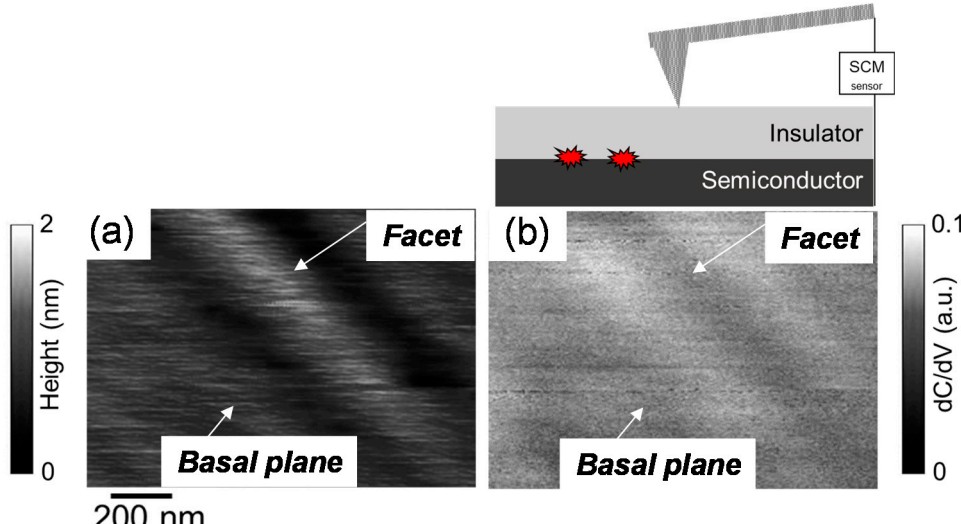

**Figure 6.** (**a**) Surface morphology acquired with an AFM scan on the bare $SiO_2$ surface, and (**b**) SCM map indicating a non-uniform spatial distribution of the surface potential.

The non-uniform spatial distribution of the interface states can be either intrinsic properties of the material—i.e., due to the different local electronic structure of the 4H-SiC crystallographic orientations—or related to a different incorporation of nitrogen during the post oxidation annealing. This aspect will be discussed more in detail in the following section.

### 3.2. Effects of Counter Doping and Interface Stress

As specified in Section 2, in order to improve the channel mobility in $SiO_2$/4H-SiC MOS-based devices, post-deposition (PDA) or post-oxidation annealing (POA) processes of the gate oxide in $N_2O$ or NO are employed. During these nitridation annealings, nitrogen can diffuse through the oxide and reach the $SiO_2$/4H-SiC interface, where it passivates the interface states. For long time, nitrogen was believed to determine only the electrical passivation of the interface states $D_{it}$. However, in 2011 Umeda et al. [15] and Kosugi et al. [49] suggested that nitrogen atoms are also incorporated in the crystalline structure of the 4H-SiC substrate, introducing shallow donor levels responsible for the increased conductivity of the MOSFET channel upon nitridation. These works used low temperature (20 K) electrically detected magnetic resonance (EDMR) to separate the effects of the nitrogen interface incorporation, i.e., interface state density reduction and dopant incorporation.

Later, Swanson et al. [50], using scanning spreading resistance microscopy (SSRM), was able to demonstrate that the nitridation process induces a "counter doping" effect of nitrogen in the p-type body region of a MOSFET. This effect has been quantified by Fiorenza et al. [51], by means of cross sectional scanning capacitance microscopy (SCM) measurements on the $SiO_2$/4H-SiC interface. In particular, SCM in cross sections showed that the faceted 4H-SiC surface morphology incorporates a larger nitrogen amount compared to the basal planes, because it exposes different ratios between (0001) and (11−20) planes [47]. Other studies based on transmission electron microscopy (TEM) [52,53] and X-ray photoemission spectroscopy (XPS) [49] demonstrated that nitrogen is incorporated within a couple of 4H-SiC crystalline monolayers.

The correlation of SCM and SSRM analyses in cross sectional samples exposed to different post oxidation annealing allowed the quantification of the counter doping effect and to evaluate the thickness of the electrically modified region underneath the $SiO_2$/4H-SiC interface. In fact, the SCM is able to quantify the doping by the comparison of the known signal collected on the epitaxial layer and on the substrate bulk [54]. On the other hand, SSRM is able to measure the local spreading resistance that is proportional to the local resistivity. Hence, SSRM is not affected by any profile broadening due to the typical depletion contribution in capacitance-based measurements (SCM).

Figure 7 shows the comparison between the conductivity profile on the as deposited (black line) and N$_2$O annealed (blue line) samples, determined using the SSRM. Fabrication details can be found in Reference [55]. As can be seen, the number of free carriers in the nitridated sample is increased by more than one order of magnitude in a region about 10 nm wide from the SiO$_2$/4H-SiC interface [47,51]. Thus, it can be concluded that the nitridation process modifies only a small fraction of the 4H-SiC crystal (one-two monolayers) but it increases the free carrier concentration in the MOSFET inversion region, effectively reducing the channel resistivity. The free carrier profile is locally increased from the doping level of the epilayer ($10^{16}$ cm$^{-3}$) to ~$10^{17}$ cm$^{-3}$, due to the nitrogen electrically active incorporation.

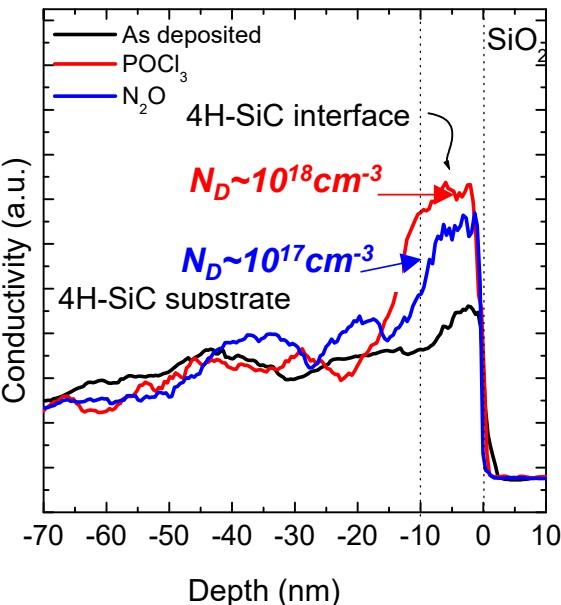

**Figure 7.** SSRM (scanning spreading resistance microscopy) carrier concentration across the SiO$_2$/4H-SiC interfaces on the as deposited (black line) and N$_2$O annealed (blue line) samples.

However, nitridation is not the only process inducing a counter doping effect. In fact, several processes have been reported in the last decade which improve the MOSFET channel resistivity. Some of them are based on the incorporation of elements of the V group of the periodic table, P [51,56–58], Sb [59], As [60], which can produce a counter doping effect similar to N atoms. As an example, Figure 7 shows also the free carrier profile on the POCl$_3$ annealed (red line) sample, collected using the SSRM. In this case, the free carrier concentration is locally increased up to ~$10^{18}$ cm$^{-3}$, thus demonstrating a higher counter doping effect induced by phosphorous compared to nitrogen.

On the other hand, the incorporation of elements of the II and III groups of the periodic table, B [7,44], Ba [45], Ca [60], La [54], Sr [60], has been investigated to explore other possible effects that explain the increase of MOSFET channel mobility. In fact, elements of the II and III groups cannot provide donors in the channel region, similar to the V group elements. The improvement of the MOSFET field effect mobility (see Table 4) induced by the use of such gate oxide process (e.g., B, Ba, Ca, Sr, or Sb at the interface) could be clarified using advanced TEM analyses, and was attributed to strain relaxation of the SiO$_2$/4H-SiC interfaces [61,62].

**Table 4.** Processes alternative to nitridation to reduce the $D_{it}$ and increase the MOSFET channel mobility.

| Gate Oxide Interface Contamination | Temperature (°C) | $\mu$ (cm$^2$V$^{-1}$s$^{-1}$) | Element Group | Counter Doping | $D_{it}$ (cm$^{-2}$eV$^{-1}$) | Ref. |
|---|---|---|---|---|---|---|
| *Lanthanum* | NA | 133 | Lanthanides | N.A. | NA | [63] |
| *Boron* | 950 | 100 | III | No | $9 \times 10^{10}$ | [61,64] |
| *Phosphorous* | 1000 | 108 | V | Yes | $5 \times 10^{11}$ | [51] |
| *Antimony* | 1150 | 65–110 | V | Yes | NA | [59] |
| *Barium* | 950 | 85 | II | No | $3 \times 10^{11}$ | [60] |
| *Calcium* | 950 | 1–5 | II | No | NA | [60] |
| *Strontium* | 950 | 40 | II | No | $3 \times 10^{11}$ | [60] |

SiO$_2$ and 4H-SiC have different thermal properties and in particular different temperature expansion coefficients. Li et al. [64] demonstrated that during the thermal growth of SiO$_2$ onto the 4H-SiC crystalline structure, the grown oxide layer creates a compressive stress along the interface. Figure 8 schematically reports on the formation of the interfacial stress during the thermal growth of the SiO$_2$ layer onto the 4H-SiC. The compressive stress (Figure 8a) produces a reduction of the average atom folding distance, which induces a reduction of the interface channel mobility of the MOSFETs. On the other hand, once the compressive stress is released (Figure 8b), the atoms' distance is increased, thus leading to an increase of the channel mobility of the MOSFETs. In fact, Huston Dycus et al. [62] demonstrated that NO annealing maintained compressive stress at the SiO$_2$/4H-SiC interface. On the other hand, the incorporation of barium atoms resulted in a release of the stress at the SiO$_2$/4H-SiC interface, producing an increase of the MOSFET channel mobility—up to $\mu_{FE}$ ~85 cm$^2$V$^{-1}$s$^{-1}$ [65].

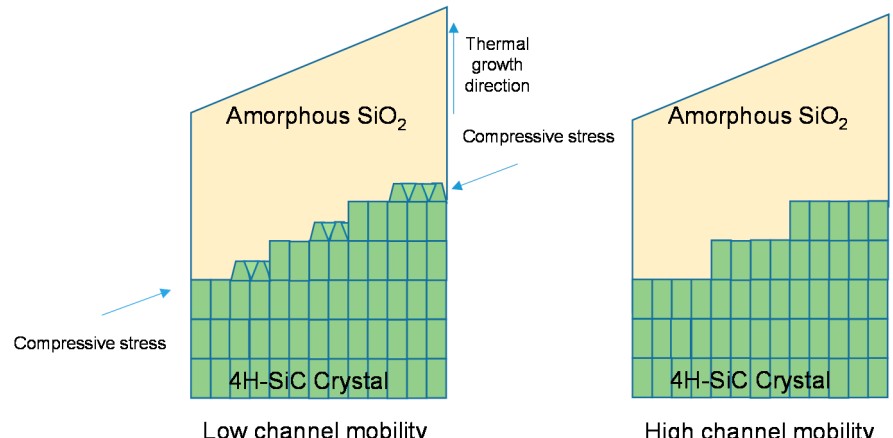

**Figure 8.** Schematic of a stressed (**a**) and a relaxed (**b**) SiO$_2$/4H-SiC interface. The thermal growth produces a compressive stress at the SiO$_2$/4H-SiC interface that limit the MOSFET channel mobility. When the interfacial stress is released (e., the MOSFET channel mobility increases.

Although the SiO$_2$/4H-SiC interface stress release by the incorporation of some foreign atoms seems to be promising to increase the 4H-SiC MOSFET channel mobility, most of these treatments exhibit some concerns related to the insulator reliability. In particular, the critical electric field and/or the conduction mechanisms are far from the ideal behaviour [66]. As an example, it has been demonstrated that P-atoms tends to form a phosphor-silicate glass (PSG) that suffers from an electron trapping, producing a pronounced $V_{th}$ instability in the MOSFETs [66].

For this reason, the nitridation process remains the preferred solution to process the SiO$_2$/SiC channel in 4H-SiC MOSFETs.

## 4. Threshold Voltage (V$_{th}$) Instability

### *4.1. Charge Trapping Phenomena*

In recent years, the threshold voltage $V_{th}$ stability in 4H-SiC MOSFETs has been a widely discussed topic, due to the relevant implications in the device performance. The $V_{th}$ stability is associated with the charge trapping phenomena occurring at the gate oxide, and it can be mitigated by the appropriate POA or PDA processes.

Figure 9 shows the degradation of the $\Delta V_{th}$ measured at 175 °C as a function of both positive and negative ($V_G = \pm 15$ V) gate bias stress time on commercial state-of-the-art 4H-SiC MOSFETs from two different generations (Gen1 and Gen2) [10]. Aviñó Salvadó et al. [67] explained the large $V_{th}$ instability that affected the Gen1 MOSFETs with an irreversible degradation of the p-n body diode, probably ascribable to a degradation of the metallization. Although the device behaviour under stress has been significantly improved in the latest commercial devices (Gen2, in Figure 9), the physical mechanisms associated to the $V_{th}$ instabilities in 4H-SiC MOSFETs remain still under debate.

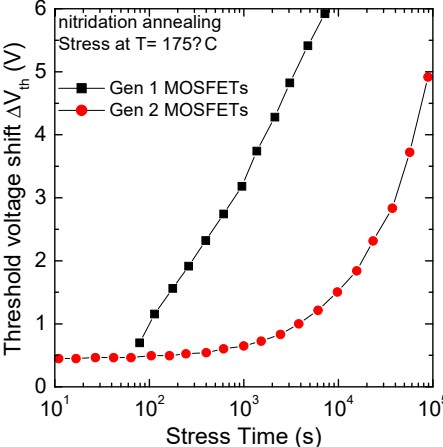

**Figure 9.** Vth shift as a function of the gate bias stress time at 175 °C for two different device families: Gen1 and Gen2 commercial 4H-SiC MOSFETs. The data are taken from Reference [10].

Interesting insights on charge trapping at the SiO$_2$/4H-SiC interface can be gained by performing a gate bias stress loop in a lateral MOSFET, as shown in Figure 10a [68]. In particular, the plot shows the $V_{th}$ as a function of the gate bias in a narrow gate bias $V_G$ range. The $V_{th}$ was determined from the linear fit of the plot of the square root of the saturation drain current $I_{DSAT}^{0.5}$, as a function of the gate bias $V_G$. Specifically, the $V_{th}$ values reported in Figure 10a were extracted by stressing the devices at a constant $V_G$ for 60 s. As can be seen, starting from the initial value $V_{th} = 8.75$ V, measured in the absence of stress (green triangle), applying an increasing positive gate bias stress induces an increase of $V_{th}$ up to 9.25 V for a stress of +20 V. Then, the $V_{th}$ measured backwards exhibits only a limited variation and $V_G$ becomes negative (i.e., $V_{th}$ remains almost stable at 9.2 V). When $V_G$ is varied from 0 to –10 V, a larger $V_{th}$ variation is observed, down to 8.25 V. This variation $\Delta V_{th}$ corresponds to $2.9 \times 10^{12}$ cm$^{-2}$ trapped electrons in the MOS system. Closing the stress loop toward the initial position ($V_G = 0$ V), the final threshold voltage value $V_{th} = 8.65$ V (red square) is still slightly lower than the initial value of 8.75 V. The same effect manifests itself, with the same amount of interface traps, through the hysteresis observed in the *C–V* measurements collected on the MOSFETs (Figure 10b) [69]. Figure 10b shows the *C–V* curves collected on a MOSFET, shorting all the terminals (body, source, and drain) and modulating the gate. Sweeping the gate bias from accumulation to inversion and backward the curves show the presence of a hysteresis due to the trapping and de-trapping of the free carriers at the interface states $N_{it} = 2.9 \times 10^{12}$ cm$^{-2}$ [69], similar to the amount of the trapped charge described in Figure 10a.

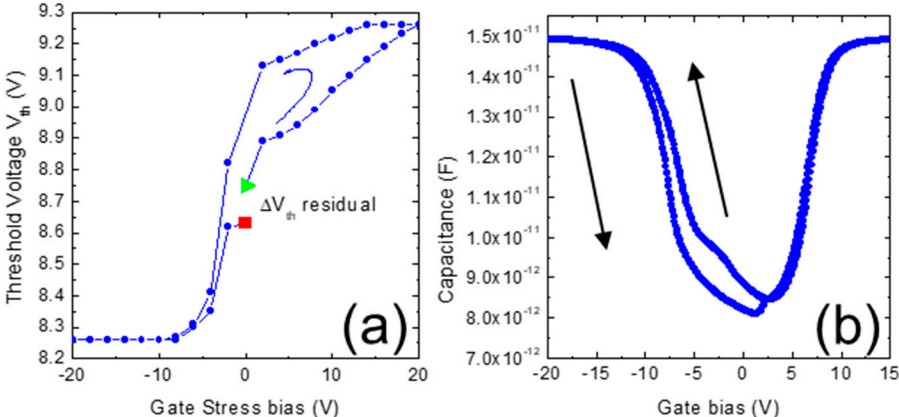

**Figure 10.** (**a**) Threshold voltage ($V_{th}$) variation measured in a lateral 4H-SiC MOSFET after a gate stress bias loop. The arrow indicates the direction of the applied stress bias loop. (**b**) Inversion-to-accumulation and backward 1 kHz $C$–$V$s of MOSFETs in gate-controlled diode configuration. The figure is adapted from References [52,68].

The residual difference between the starting and the ending point of the stress loop shown in Figure 10a must be correlated to traps slower than the interface states. This difference $\Delta V_{th} = 0.1$ V can be explained by the presence of a residual charge density of $1.5 \times 10^{11}$ cm$^{-2}$, which remains trapped in the system and needs several hours to recover and go back at the original condition. This residual charge is associated with the presence of near interface traps (NIOTs) within the insulator in the SiO$_2$/4H-SiC system (see Figure 3). However, the behaviour described in Figure 10, often observed in 4H-SiC MOSFETs, is still under debate and deserves further attention due to its possible implications during high-frequency (>100 kHz) switching operations.

In general, slow NIOTs and bulk traps in the insulator are believed to be responsible for the observed minor $V_{th}$ instability effects. In principle, due to their long relaxation time, NIOTs can be detected using MOSFET gate current measurements. However, the broad time response range of the trapping states make their detection complicated. In particular, NIOTs close to the SiO$_2$/4H-SiC interface are much faster than those that are located deeper into the bulk of the insulator. Hence, slow characterization methods are unable to monitor fast states, and conversely, slow traps need a long stress time to be stimulated.

In literature, different approaches have been used to address these difficulties. The scenario is summarized in the following part.

During a gate bias stress, a $V_{th}$ shift occurs either due to a charge injection in the gate oxide or the generation of interface states. The filled traps tend to come back to the equilibrium after the bias is supressed. Hence, the total amount of charges in the NIOTs starts to decrease after the removal of the gate bias and the $V_{th}$ shift reflects the number of residual still trapped charges during the subsequent measurements. To accurately determine the traps, a number of high-speed characterization methods are required.

Many studies reported on the $V_{th}$ instability upon negative gate bias stress on MOSFETs [9,52,70]. However, to get insights on the basic trapping mechanisms of NIOTs in the insulator, it is useful to analyse the behaviour of p-type MOS capacitors (Figure 11) [71].

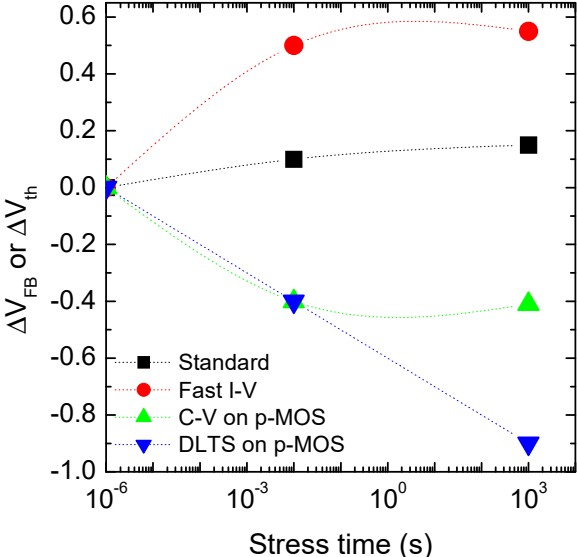

**Figure 11.** $\Delta V_{FB}$ (equivalent to $\Delta V_{th}$) as a function of the stress time measured with different electrical characterization techniques. Data taken from Reference [71].

Figure 11 shows a comparison of different techniques to determine the shift $\Delta V_{th}$ in MOSFETs (current measurements) and the flat band voltage shift $\Delta V_{FB}$ in MOS capacitors (capacitance measurements). In Figure 11, the "standard" $\Delta V_{th}$ measurement was obtained with a full $I_{DS}$–$V_{GS}$ transcharacteristic, i.e., sweeping the gate bias from negative to large positive values (changing the polarization direction). In the fast $I$–$V$ method, the MOSFET drain current $I_D$ was measured at $V_G$ = 6.5, 7.0, and 7.5 V, and the $\Delta V_{th}$ values were estimated from the shift of the curves at constant $I_{DS}$. In the $C$–$V$ method, the $C$–$V$ curves on p-type MOS capacitors were obtained in a range from $-15$ to $-3$ V. Then, $\Delta V_{FB}$ values were estimated from the shift of the $C$–$V$ curves at constant capacitance in the depletion region, taking 10 s to collect the information. Traps with a relaxation time shorter than the measurement time are not detectable. In the CC-DLTS method (constant capacitance deep level transient spectroscopy), the gate voltages of the constant capacitance were measured with a time resolution of 10 ms. In particular, after the gate bias stress, the properly designed measurement circuit allowed the monitoring of the gate bias needed to obtain the fixed capacitance value. Then, the gate voltage shift was regarded as $\Delta V_{FB}$. Thus, the DLTS method allows the detection of traps with a relaxation time four decades shorter than standard methods.

As can be noticed in Figure 11, the flat band voltage shift $\Delta V_{FB}$ determined by standard $C$–$V$ measurements ($\Delta V_{FB} \sim -0.4$ V) by sweeping the $V_G$ in a wide range is comparable to that measured on MOSFETs ($I_D$–$V_G$). On the other hand, a deep-level transient spectroscopy (DLTS) method has been used to perform fast $C$–$V$ measurements at constant gate capacitance. Hence, thanks to the reduction of the time needed to measure the $V_{th}$ shift, it was possible to better estimate the amount of NIOTs. In fact, after the same stress time (i.e., 1000 s at $V_G = -15$ V), the $\Delta V_{th}$ estimated with fast $C$–$V$ measurements is more than doubled compared with that measured with standard $C$–$V$ measurements. This can be understood considering that the charge from the semiconductor substrate needs time to access the NIOTs via tunnelling. In particular, according to the Wentzel–Kramers–Brillouin approximation, a single tunnelling happens at a certain distance in logarithmic time. Bauza and Ghibaudo [72] have presented a single tunnelling model to describe the trapping phenomena in silicon MOS capacitors. However, interface traps contribute to the trapping phenomena due to the increased tunnelling probability. In 4H-SiC MOSFETs the $D_{it}$ is two orders of magnitude larger than in silicon. Hence, $D_{it}$ must be taken into account to study the NIOTs trapping phenome. According to Paulsen et al [73],

the $\tau(x)$ tunnelling times needed to reach the NIOTs from the 4H-SiC substrate can be calculated as a function of the distance from the interface toward the bulk of the insulator, according to:

$$\tau(x) = \frac{m_1^* x \left(1 + \frac{1}{2\eta_1 x}\right)}{2\pi \eta_2 \hbar^3 D_{it}} exp(2\eta_1 x) \tag{5}$$

where $\eta_1$ and $\eta_2$ are functions of the doping of the material, and where $m_1{}^*$ is the effective mass for electrons in the oxide [73]. Furthermore, $\hbar$ is the reduced Plank constant. Using the literature values of $m_1{}^*$ [74], Equation (5) can be drawn as a function of the distance x between the NIOTs and the SiO$_2$/4H-SiC interface. Thus, the tunnelling time constant as a function of the NIOTs distance and the SiO$_2$/4H-SiC interface is shown in Figure 12.

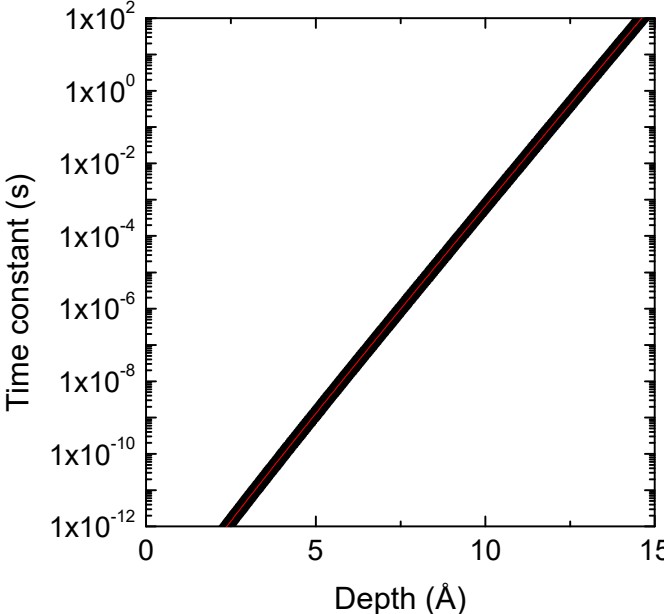

**Figure 12.** Tunneling time constant obtained varying the depth between the near interface oxide traps (NIOTs) and the SiO$_2$/4H-SiC interface. Data calculated from References [52,73] using Equation (5).

Clearly, a long stress time is needed to charge NIOTs located far away from the SiO$_2$/4H-SiC interface. In order to freeze the charged NIOTs, it is preferable to avoid the interruption of the gate bias and to reduce as much as possible the time needed to perform the $V_{th}$ shift measurement. In the last decade, several time-resolved capacitance- and current-measurements have been employed to investigate the NIOTs [13,14,75]. A faster time resolution provides more information on NIOTs close to the SiO$_2$/4H-SiC interface. Recently Pande et al [76] presented a method to investigate NIOTs on MOS capacitors in strong accumulation based on AC current measurements, with a time resolution of a few tenths of a nanosecond ($\sim 10^{-8}$ s). With this method, an amount of NIOTs in the order of $10^{13}$ cm$^{-2}$ has been measured. Clearly, looking at Figure 12, with a time resolution of $10^{-8}$ s the NIOTs located within the first 5 Å from SiO$_2$/4H-SiC interface results are unmeasurable.

## 4.2. Chemical Nature of the NIOTs

The chemical nature of the NIOTs has been theoretically debated [77,78]. In particular, C-C dimers and interstitial silicon atoms create both interface and near interface states at the SiO$_2$/SiC interfaces. However, a clear identification and unification of the different processes is not yet provided. In order to gain deep insight into the chemical nature of the NIOTs responsible for the $V_{th}$ instability, and on the near SiO$_2$/4H-SiC interface region, several microscopic chemical investigations have been presented in literature in the last decade [52,53,79,80]. The different results reported in literature are

often due to the variety of $SiO_2$/SiC interface processing. In particular, first Zheleva et al. [79] and then Biggerstaff et al. [80] have investigated the $SiO_2$/SiC interface at nanoscale, employing transmission electron microscopy (TEM). They detected the presence of a transition layer (up to several nanometers thick) containing carbon atoms at the $SiO_2$/4H-SiC interface in thermally grown oxides onto 4H-SiC. However, the last generation of thermally grown oxides is much different than those pioneering works. In fact, Regoutz et al. [53], using X-ray photoemission spectroscopy (XPS), have recently demonstrated the presence of an amount of N–C–Si bonds at the interface, which depends on the different nitridation processes. Clearly, the $SiO_2$/SiC interfaces described in the literature are often difficult to compare, since different processes are used for the $SiO_2$ deposition/growth or for the passivation of the interface states.

Recently, Fiorenza et al. [52] have reported on the presence of a narrow sub-stoichiometric $SiO_x$ layer produced on the re-oxidation of the 4H-SiC surface, even during nitridation (i.e., NO and $N_2O$ thermal treatments can move the $SiO_2$/4H-SiC oxidation interface) of a deposited oxide layer. In particular, sub-Ångström resolution scanning transmission electron microscopy (STEM) analyses combined with electron energy loss spectroscopy (EELS) have been used to monitor the chemical environment of the first nanometer of insulator from the 4H-SiC interface. Furthermore, the nano-chemical investigation was correlated with the time-dependent capacitance measurements and correlated with the NIOTs responsible for the MOSFET threshold voltage instability [52].

Figure 13a shows the cross-section of the $SiO_2$/4H-SiC interface in high resolution dark field (DF) spectrum image that simultaneously collected all the elemental EELS maps. The chemical element profiling a sub-stoichiometric $SiO_x$ layer was obtained from the Silicon map using a 4 eV wide energy window between 99 and 103 eV (Figure 13b). In fact, in this 4 eV wide energy window, only the silicon atoms not completely oxidized (surrounded by four oxygen atoms) can give a contribution to the EELS spectrum above the noise limit. Figure 13b highlights the progressive change in the $SiO_x$ and oxygen profiles across the $SiO_2$/4H-SiC interface. This experimental evidence suggests the presence of a non-abrupt (NA) $SiO_2$/4H-SiC interface. Furthermore, the carbon profile decreases within the oxide, with a tail that is wider than one nanometre.

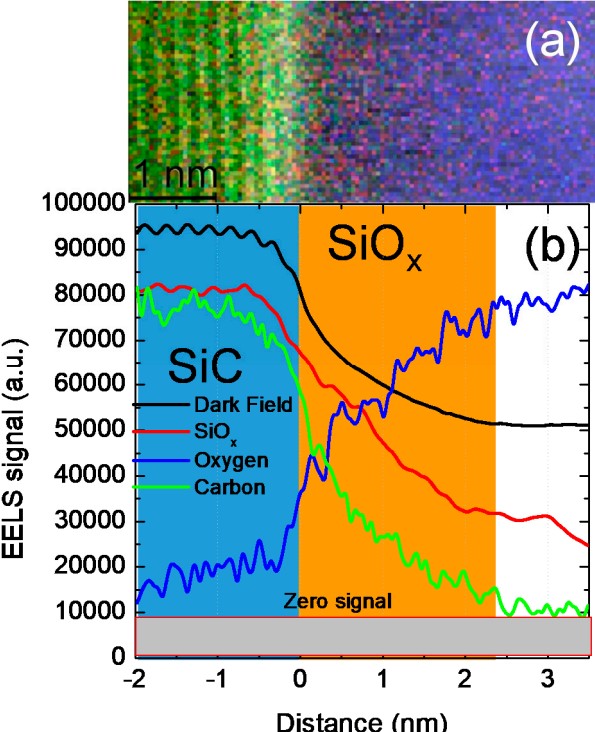

**Figure 13.** (**a**) High-resolution spectrum imaging cross section STEM image of the $SiO_2$/4H-SiC interface. (**b**) EELS spectra of carbon, oxygen, and partially oxidized silicon ($SiO_x$). More details on these analyses are reported in Reference [52].

In this scenario, clearly further investigations are needed to correlate the $V_{th}$ instability with particular NIOTs and to identify their chemical nature. Then, an appropriate technological strategy to overcome this issue has to be pursued to further improve the performances and the reliability concerns of commercial 4H-SiC power MOSFETs.

## 5. Conclusions

This review paper presented some relevant characterization aspects of the $SiO_2$/SiC system in 4H-SiC MOSFETs. A special emphasis is given to the need to correlate several standard and nanoscale electro-structural techniques, in order to have an exhaustive scenario of the $SiO_2$/4H-SiC properties and a better comprehension of the 4H-SiC MOSFET physics in relation to the device's processing steps. In particular, it has been shown that NIOTs have a strong influence on the device's electrical behaviour. The cross correlation of time dependent electrical measurements with structural analyses enabled the estimatation of a typical amount of NIOTs of $10^{11}$ cm$^{-2}$ inside the $SiO_2$ (within 1–2 nm from the SiC interface) and attribute them to the presence of a sub-stoichiometric $SiO_x$ layer at the interface. Hence, to characterize these interfaces, advanced time resolved fast electrical measurements are mandatory to discriminate between different traps at the interface and within the oxide.

**Author Contributions:** Conceptualization, Writing-Original Draft Preparation, P.F.; Writing-Review & Editing, Supervision, Project Administration, F.R.; Methodology, Investigation, F.G.

**Funding:** This work was partially carried out in the framework of European project WInSiC4AP (Wide Band Gap Innovative SiC for Advanced Power), funded by the ECSEL JU under grant agreement No.737483. This Joint Undertaking receives support from the European Union's Horizon 2020 research and innovation programme and Czech Republic, France, Germany, Italy. The WInSiC4AP project is also supported by ESI funds from MIUR 2014-2020 FESR program.

**Acknowledgments:** The authors would like to acknowledge M. Saggio and F. Iucolano (STMicroelectronics) and their teams for the collaboration. In particular, A. Parisi and S. Reina are acknowledged for support during electrical characterization. Moreover, the authors thank all their colleagues of CNR-IMM. In particular, A. La Magna and I. Deretzis for support with theoretical calculations, G. Nicotra and C. Bongiono for TEM investigation, S. Di Franco for devices processing, R. Lo Nigro, G. Greco and E. Schilirò for fruitful discussions.

**Conflicts of Interest:** The authors declare no conflict of interest.

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
