# Peer review of "Characterization of SiO2/4H-SiC Interfaces in 4H-SiC MOSFETs: A Review"

_energies, doi:10.3390/en12122310_

Round 1
Reviewer 1 Report
No comment
Author Response
We would like to thank the reviewers for their valuable comments and suggestions that allowed us to improve the overall quality of this review paper. We hope that the revised manuscript is now suitable for publication on Energies.
Below you can find a response for each reviewers’ comment.
#1 Comments and Suggestions for Authors
This manuscript is a good review of state-of-the-art characterization of the SiO2/4H-SiC interface that is a key technique to improve the power MOSFET.
The text and the references contained will be valuable for anyone venturing into this research field.
I have found some minor errors that can be easily corrected (end of review) but I would also like to suggest a few additional references, and the mention of one more technique that I found was perhaps overlooked.
Charge Pumping
Although charge-pumping is perhaps not the most commonly used in SiC interface characterization, it surely is in Si technology.
The authors should list this technique in table II (see refs provided) and then either explain why it is not as useful, or why another technique can be used.
Charge pumping refs, SiC:
Okamoto, 10.1109/TED.2008.926639
Martin, 10.4028/www.scientific.net/MSF.740-742.891,
Salinaro, 10.1109/TED.2014.2372874
We thank the reviewer for the comment on the charge pumping. We have included the relevant references in the revised manuscript. Furthermore, the technique is now included also in Table II. In particular, in the manuscript the following part has been added:
“Another characterization technique, the charge pumping method, commonly used in silicon devices [ ], is recently rising relevance also for SiC devices. The method is based on measuring the transistor base charge-pumping current while applying voltage pulses (variable amplitudes and lengths) to its gate. In particular, the difficulty to interpret anomalous data collected on 4H-SiC MOSFETs has been firstly explained by Okamoto et al [ ] in terms of geometric component and the acceptor-like interface states. In fact, due to the low channel mobility, the charge carriers need a certain time to cross the channel. Hence, an accurate setting of the experimental procedure is needed. More recently, Salinaro et al [ ] found an appropriate temperature, frequency and bias amplitude to demonstrate not only the interface state distribution but also a not homogeneous doping distribution in the channel due to the device processing.”
Tunneling
The model in Equation (5) is a special case with tunneling from IT to NIT. There are other models involving single step tunneling to NIT, see
Bauza och Ghibaudo, 10.1016/0038-1101(95)00156-5
The important message that should be emphasized is that tunneling happens in logarithmic time,
xm = ln(t/tau)/2 kappa , where kappa is the attenuation coefficient from the WKB approximation for tunneling.
We thank the reviewer for the comment on the tunnelling mechanism. We have included the reference and we emphasized the difference between the two different models. The tunnelling section of the manuscript has been amended as follows:
“In particular, according with the Wentzel-Kramers-Brillouin approximation, a single tunnelling happens at a certain distance in logarithmic time. Bauza and Ghibaudo [ ] have presented a single tunnelling model to describe the trapping phenomena in silicon MOS capacitors. However, interface traps contribute to the trapping phenomena due to the increased tunnelling probability. In 4H-SiC MOSFETs the Dit is two orders of magnitude larger than in silicon. Hence, Dit must be taken into account to study the NIOTs trapping phenome.”
Figure 1 is based on data from ref [2] from 2012. Are there any newer results that can be added?
Unfortunately, based on our knowledge, since the SiC MOSFETs are available on market the manufactures are not publishing homogeneous data directly comparable with those obtained at research level. We hope that the reviewer can understand our difficulties to address this point.
Figure clarity
Figure 2 is a bit dark and it can perhaps be clarified with some resistor symbols superimposed?
Figure 2 has been modified accordingly with the reviewer’s comment
The JFET resistance can perhaps be explained, it is not clear from this figure what causes it.
Equation 2
Motivate why inversion mobility is the key, and why not channel length, pitch or Cox.
The clarification concerning the JFET region and the explanation of the terms included in the Eq describing the channel resistance have been unified and included in the manuscript as follows:
“…where p is the pitch of the MOSFET elementary cell, Lch is the channel length, µinv is the mobility for electrons in the channel (inversion layer), Cox is the specific capacitance of the gate oxide, Vth is the threshold voltage, and VG is the applied gate bias. It is important to emphasize that power devices must have a good trade-off between the ON- and OFF-state. In particular, for a given breakdown voltage and threshold voltage the choice of a drift layer and body region with a fixed doping levels and thicknesses are required. Hence, the minimum pitch size is obtained avoiding the overlap of the depleted region of the body-drain p-n diode (in the JFET region). Furthermore, the Rch can be influenced by the Cox. However, to guarantee the reliability of power devices, insulators with high critical breakdown field, large band gap and high melting point are desired. So far, the most robust gate insulation solution is the SiO2, but the recent literature is exploring alternative solutions such as Al2O3 [ ] and AlON [ ]. Moreover, the channel resistance Rch can be lowered by reducing the channel length. However, when the MOSFET operates in interdiction and it is reverse biased short-channel phenomena have to be avoided.”
Equation 3
(L, W and Cox are already defined)
They have been suppressed.
Table I
Column heading "Dit or Nit" can perhaps be labeled extraction method instead?
Table I has been modified
Table II
Conductance method is the same as Gp/w in Table I, right? Don't mix names.
The nomenclature has been unified
The conductance method under-estimates the trap density if q^2 Dit > Cox, which is commonly the case in literature.
See Martens et al, 10.1109/TED.2007.912365
The comment has been included in the revised manuscript: “Hence, the above mentioned techniques underestimate the distribution of Dit when q2·Dit > Cox [[i]]. Furthermore, the Gp/ω method is not sensitive to the fast states that can be undetectable in the usually employed frequency range (up to few MHz; see Table II)”
Line 178-179 "...is difficult to measure accurately..." Mention that split-CV [33] is commonly used?
The split-CV is mentioned: “In fact, to accurately determine the Qn complicate experimental setups are needed in order to perform the split C-V measurements [35] separating the source-drain and gate-base capacitance response.”
Line 213-214 "...increasing with 1/Nit." Can this sentence be rewritten, it confused me at first?
The sentence has been rewritten: “Clearly, the values of the mobility increases decreasing the amount of Nit.”
Line 316 "...as deposited..." Yes but what was deposited? Some more experimental details or references please.
The sentence has been modified: “Fabrication details can be found in Ref. [[ii]].”
Section 4.2 Chemical nature of the NIOTs
Perhaps theoretical studies of oxide defects in SiO2/SiC can be mentioned, for instance
Knaup et al, 10.1103/PhysRevB.72.115323 and
Devynck et al, 10.1103/PhysRevB.84.235320
The theoretical studies have been mentioned in the revised manuscript: “The chemical nature of the NIOTs has been theoretically debated [[iii],[iv]]. In particular, C-C dimers and interstitial silicon atoms create both interface and near interface state at the SiO2/SiC interfaces.”
Minor errors
(Throughout)
Unit placement is sometimes next to the value like 1200V, line 30, but also lines 211, 381-389, 397, 398, and table II, figure caption 10.
Degrees C should be written without spaces before and after degree sign, line 65, 134-138, 276, maybe more.
Line 183 "relater" -> "related"
Line 205 Table III caption is incomplete?
In table III third row "Field effective mobility" -> "Field effect mobility"
Line 224 "interface sates" -> "interface states"
Line 257 What does 10 divided by 100 kHz mean? Should there be a tilde or dash instead?
Line 260 attoF is written just aF
Line 267 missing a final "."
Line 308 "...of4H-SiC..." missing a space
Amorphous is misspelled in Fig. 8.
Line 455 "... NIOTs .located..." an extra period before "located"
Line 496 "...it results clear..." please rewrite
All the mentioned typing errors have been fixed.
[i] Martens, K.; On Chui, C.; Brammertz, G.; De Jaeger, B.; Kuzum, D.; Meuris, M.; Heyns, M.M.; Krishnamohan, T.; Saraswat, K.; Maes, H.E.; Groeseneken, G. On the Correct Extraction of Interface Trap Density of MOS Devices With High-Mobility Semiconductor Substrates. IEEE Trans. Electron. Dev. 2008, 55, 547
[ii] Fiorenza, P. ; Vivona, M. ; Iucolano, F. ; Severino, A. ; Lorenti, S. ; Nicotra, G. ; Bongiorno, C. ; Giannazzo, F. ; Roccaforte, F. Temperature-dependent Fowler-Nordheim electron barrier height in SiO2/4H-SiC MOS capacitors. Mater. Sci. Semicon. Processing 2018 78 38–42
[iii] Knaup, J. M.; Deák, P.; Frauenheim, Th.; Gali, A.; Hajnal, Z.; Choyke, W. J. Defects in SiO2 as the possible origin of near interface traps in the SiC/SiO2 system: A systematic theoretical study Phys. Rev. B 2005 72, 115323
[iv] Devynck, F.; Alkauskas, A.; Broqvist, P.; Pasquarello, A. Charge transition levels of carbon-, oxygen-, and hydrogen-related defects at the SiC/SiO2 interface through hybrid functionals Phys. Rev. B 2011 84, 235320

Reviewer 2 Report
This manuscript is a good review of state-of-the-art characterization of the SiO2/4H-SiC interface that is a key technique to improve the power MOSFET.
The text and the references contained will be valuable for anyone venturing into this research field.
I have found some minor errors that can be easily corrected (end of review) but I would also like to suggest a few additional references, and the mention of one more technique that I found was perhaps overlooked.
Charge Pumping
Although charge-pumping is perhaps not the most commonly used in SiC interface characterization, it surely is in Si technology.
The authors should list this technique in table II (see refs provided) and then either explain why it is not as useful, or why another technique can be used.
Charge pumping refs, SiC:
Okamoto, 10.1109/TED.2008.926639
Martin, 10.4028/www.scientific.net/MSF.740-742.891,
Salinaro, 10.1109/TED.2014.2372874
Tunneling
The model in Equation (5) is a special case with tunneling from IT to NIT. There are other models involving single step tunneling to NIT, see
Bauza och Ghibaudo, 10.1016/0038-1101(95)00156-5
The important message that should be emphasized is that tunneling happens in logarithmic time,
xm = ln(t/tau)/2 kappa , where kappa is the attenuation coefficient from the WKB approximation for tunneling.
Recent data?
Figure 1 is based on data from ref [2] from 2012. Are there any newer results that can be added?
Figure clarity
Figure 2 is a bit dark and it can perhaps be clarified with some resistor symbols superimposed?
The JFET resistance can perhaps be explained, it is not clear from this figure what causes it.
Equation 2
Motivate why inversion mobility is the key, and why not channel length, pitch or Cox.
Equation 3
(L, W and Cox are already defined)
Table I
Column heading "Dit or Nit" can perhaps be labeled extraction method instead?
Table II
Conductance method is the same as Gp/w in Table I, right? Don't mix names.
The conductance method under-estimates the trap density if q^2 Dit > Cox, which is commonly the case in literature.
See Martens et al, 10.1109/TED.2007.912365
Line 178-179 "...is difficult to measure accurately..." Mention that split-CV [33] is commonly used?
Line 213-214 "...increasing with 1/Nit." Can this sentence be rewritten, it confused me at first?
Line 316 "...as deposited..." Yes but what was deposited? Some more experimental details or references please.
Section 4.2 Chemical nature of the NIOTs
Perhaps theoretical studies of oxide defects in SiO2/SiC can be mentioned, for instance
Knaup et al, 10.1103/PhysRevB.72.115323 and
Devynck et al, 10.1103/PhysRevB.84.235320
Minor errors
(Throughout)
Unit placement is sometimes next to the value like 1200V, line 30, but also lines 211, 381-389, 397, 398, and table II, figure caption 10.
Degrees C should be written without spaces before and after degree sign, line 65, 134-138, 276, maybe more.
Line 183 "relater" -> "related"
Line 205 Table III caption is incomplete?
In table III third row "Field effective mobility" -> "Field effect mobility"
Line 224 "interface sates" -> "interface states"
Line 257 What does 10 divided by 100 kHz mean? Should there be a tilde or dash instead?
Line 260 attoF is written just aF
Line 267 missing a final "."
Line 308 "...of4H-SiC..." missing a space
Amorphous is misspelled in Fig. 8.
Line 455 "... NIOTs .located..." an extra period before "located"
Line 496 "...it results clear..." please rewrite
Author Response

(The authors gave the same response as above.)

Reviewer 3 Report
Characterization of SiO2/4H-SiC interfaces in 4H-SiC MOSFETs: A review
This paper describes an overview about characterization methods of the SiO2/4H-SiC interfaces used to assess the behavior of MOSFETs. In particular, the importance of correlating conventional electrical analyses of devices and test patterns, with more advanced characterization techniques at the nanoscale is highlighted. This is well written and organized paper. It is scientifically sound and contains sufficient interest.
Reviewer’s comments:
1. In the abstract part, I fail to understand the scientific contribution of this research. What’s the scientific contribution of this research?
2. In sentences/equations, mathematical expressions must be Italic font. Unify the font style.
3. The authors explained that “Table I compares several literature data of the channel mobility measured in 4H-SiC MOSFETs, fabricated either on epitaxial or ion-implanted layers, and subjected to annealing of the gate oxides in nitrogen-rich conditions [19,20,21,22,23,24,25,26,27,28].” However, these reference number does not match to the reference quoted in Table 1.
4. In Table 1, the explanation of several parameters is missing. The meaning of these parameters is not clear.
5. In Conclusion, in-deep discussion of the findings should be given. For example, the authors explained that “Clearly, the SiO2/SiC system is still far from an ideal behaviour. Hence, further processing improvement are required, to optimize the interface states and stress.” This is a trivial matter.
Author Response
#2 Comments and Suggestions for Authors
Characterization of SiO2/4H-SiC interfaces in 4H-SiC MOSFETs: A review
This paper describes an overview about characterization methods of the SiO2/4H-SiC interfaces used to assess the behavior of MOSFETs. In particular, the importance of correlating conventional electrical analyses of devices and test patterns, with more advanced characterization techniques at the nanoscale is highlighted. This is well written and organized paper. It is scientifically sound and contains sufficient interest.
Reviewer’s comments:
1. In the abstract part, I fail to understand the scientific contribution of this research. What’s the scientific contribution of this research?
We thank the reviewer for the comment and we have modified the abstract and the conclusion (see the comment below). In the abstract it is written now: “This paper gives an overview on some state-of-the-art characterization methods of SiO2/4H-SiC interfaces in metal oxide semiconductor field effect transistors (MOSFETs). In particular, the work compares benefits and drawbacks of different techniques to assess the physical parameters describing the electronic properties and the current transport at the SiO2/SiC interfaces (interface states, channel mobility, trapping phenomena, etc.). First, the most common electrical characterization techniques of SiO2/SiC interfaces are presented (e.g., capacitance- and current-voltage techniques, transient capacitance and current measurements, etc.). Then, examples of electrical characterizations at the nanoscale (by scanning probe microscopy techniques) are given, to get insights on the homogeneity of the SiO2/SiC interface and the local interfacial doping effects occurring upon annealing. The trapping effects occurring in SiO2/4H-SiC MOS systems are elucidated using advanced capacitance and current measurements as a function of time. In particular, these measurements give information on the density (~1011cm-2) of near interface oxide traps (NIOTs) present inside the SiO2 layer and their position with respect to the interface with SiC (at about 1-2 nm). Finally, it will be shown that a comparison of the electrical data with advanced structural and chemical characterization methods allows to ascribe the NIOTs to the presence of a sub-stoichiometric SiOx layer at the interface.”
2. In sentences/equations, mathematical expressions must be Italic font. Unify the font style.
The font in the equation has been modified.
3. The authors explained that “Table I compares several literature data of the channel mobility measured in 4H-SiC MOSFETs, fabricated either on epitaxial or ion-implanted layers, and subjected to annealing of the gate oxides in nitrogen-rich conditions [19,20,21,22,23,24,25,26,27,28].” However, these reference number does not match to the reference quoted in Table 1.
Table I has been modified including all the mentioned reference in the text
4. In Table 1, the explanation of several parameters is missing. The meaning of these parameters is not clear.
Description of Table I has been clearly written.
5. In Conclusion, in-deep discussion of the findings should be given. For example, the authors explained that “Clearly, the SiO2/SiC system is still far from an ideal behaviour. Hence, further processing improvement are required, to optimize the interface states and stress.” This is a trivial matter.
The conclusion has been modified including an addition sentence that perhaps clarify the aim of the paper: “This review paper presented some relevant characterization aspects of the SiO2/SiC system in 4H-SiC MOSFETs. A special emphasis is given to the need of correlating several standard and nanoscale electro-structural techniques, in order to have an exhaustive scenario of the SiO2/4H-SiC properties and a better comprehension of the 4H-SiC MOSFET physics in relation to the devices processing steps. In particular, it has been shown that NIOTs have a strong influence on the device electrical behaviour. The cross correlation of time dependent electrical measurements with structural analyses enabled to the estimate a typical amount of NIOTs of 1011cm-2 inside the SiO2 (within 1-2 nm from the SiC interface) and attribute them to the presence of a sub-stoichiometric SiOx layer at the interface. Hence, to characterize these interfaces, advanced time resolved fast electrical measurements are mandatory to discriminate among different traps at the interface and within the oxide.”
